# STICKING TO THE FACTS: CONFIDENT DECODING FOR FAITHFUL DATA-TO-TEXT GENERATION

## ABSTRACT

Neural conditional text generation systems have achieved significant progress in recent years, showing the ability to produce highly fluent text. However, the inherent lack of controllability in these systems allows them to *hallucinate* factually incorrect phrases that are unfaithful to the source, making them often unsuitable for many real world systems that require high degrees of precision. In this work, we propose a novel confidence oriented decoder that assigns a confidence score to each target position. This score is learned in training using a variational Bayes objective, and can be leveraged at inference time using a calibration technique to promote more faithful generation. Experiments on a structured data-to-text dataset – WikiBio (Lebret et al., 2016) – show that our approach is more faithful to the source than existing state-of-the-art approaches, according to both automatic metrics and human evaluation.

## 1 INTRODUCTION

Conditional text generation is the task of generating some target text $y$ conditioned on source content $x$. It is the essence of many natural language processing problems, such as text summarization (Mani, 1999), where $x$ is a long document and $y$ is a more concise version, machine translation (Koehn, 2009), where $x$ and $y$ represent equivalent text in different languages, and data-to-text generation (Kukich, 1983; McKeown, 1992), where $x$ is a structured table and $y$ is a textual description.

While traditionally done with template-based approaches (Becker, 2002; Foster & White, 2004; Gatt & Reiter, 2009; Reiter et al., 2005), recently neural encoder-decoder approaches (Sutskever et al., 2014; Cho et al., 2014; Bahdanau et al., 2014) have become a popular approach. In this formulation, the source content is encoded with a neural architecture, and the decoder autoregressively produces a token at each output position based on its internal state and the source representation. By leveraging continuous representations with rich non-linearities, encoder-decoder approaches can generate highly fluent text (Rush et al., 2015; Radford et al., 2019) without the need for cumbersome handcrafted rules and templates.

However, encoder-decoder architectures are inherently difficult to control, and have been shown to be prone to *hallucination*, i.e., generating text that is fluent but *unfaithful* to the source (Vinyals & Le, 2015; Koehn & Knowles, 2017; Wiseman et al., 2017; Lee et al., 2018). This severe shortcoming can often limit the use of neural approaches in many real world systems, where it is not acceptable to produce output that is even occasionally unfaithful.

In this work, we focus on data-to-text generation since the structured form of source content $x$ makes it relatively easy to evaluate faithfulness using both human evaluation and domain-specific automatic metrics (Dhingra et al., 2019). In particular, we focus on the WikiBio (Lebret et al., 2016) dataset, where the task is to generate a sentence summarizing a tabular biography of a person. Figure 1 shows an example.

First note that the reference contains information such as *bonanno crime family* and *informant* that are true, but cannot be inferred from the source. This source-reference *divergence* exists in many large-scale generation datasets (Wiseman et al., 2017; Dhingra et al., 2019). Secondly, most generation systems are agnostic to this divergence and trained to maximize the log-likelihood of reference. This can often encourage the models to output phrases that are unsupported by the source. For example, Figure 1 shows the output of a state-of-the-art generation baseline, the Pointer-Generator

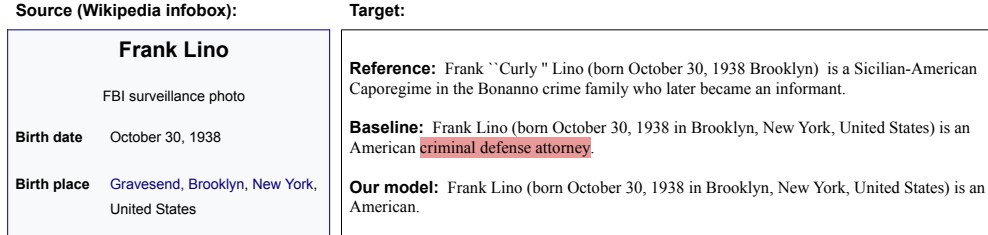

Figure 1: Example in the WikiBio dataset (Lebret et al., 2016) showing the biography of *Frank Lino*. The baseline Pointer-Generator (See et al., 2017) exhibits hallucination.

network (See et al., 2017), which contains the phrase *criminal defense attorney* that is false (but loosely related to *FBI* in the table). Thus, hallucination can often result from the coupling of model shortcomings (e.g. lack of formal reasoning, learning false correlations), and noise/artifacts in the training data.

In this work, we propose a confidence oriented approach which assigns a learned confidence score to each decoder position, and then uses the score in two ways to reduce hallucination: **(1)** In test, it uses confidence to adjust the output probabilities by a calibration technique (Braverman et al., 2019). **(2)** In training, we employ a variational Bayes objective to jointly learn the confidence score while allowing the model to skip tokens with a low confidence score to avoid training on reference phrases that are difficult to infer from the source. In Figure 1, our approach leads to a faithful generation that omits the occupation.

Empirically, when evaluated on the WikiBio dataset (Lebret et al., 2016), we show that our approach is considerably more faithful to the source than existing state-of-the-art solutions, according to both PARENT precision (Dhingra et al., 2019) and human evaluation.

## 2 RELATED WORK

Improving the fidelity and accuracy of text generation systems is an important research topic that has spawned a variety of different approaches. Some focus on blending extractive and abstractive methods, e.g., allowing the model to copy tokens directly from the source (Gu et al., 2016; See et al., 2017), separating content selection from generation (Zhou et al., 2017; Gehrmann et al., 2018) and utilizing topic information from the source to make informed generation (Narayan et al., 2018).

Other approaches have proposed generating more accurate text using semiparametric approaches (Guu et al., 2018; Pandey et al., 2018; Peng et al., 2019), reinforcement learning-based rewards (Paulus et al., 2018; Pasunuru & Bansal, 2018), semi-Markov models to learn neural templates (Wiseman et al., 2018), data augmentation (Ma et al., 2019; Kedzie & McKeown, 2019), content planning (Puduppully et al., 2019a), and constrained vocabulary decoding (Wu et al., 2017). While many leverage the structure of the source (Liu et al., 2018; Marcheggiani & Perez-Beltrachini, 2018) or task-based insights (Puduppully et al., 2019b), our approach is complementary in that it uses general machine learning techniques to build a confidence oriented decoder, that is more faithful to the source and robust to divergence/noise in the training data. Furthermore, many previous works rely on automatic metrics such as BLEU, which can be poorly correlated with human judgment of faithfulness (Wiseman et al., 2017; Dhingra et al., 2019). In contrast, we evaluate on PARENT precision (Dhingra et al., 2019), a metric specifically designed to capture faithfulness in data-to-text generation, and conduct a rigorous human evaluation to assess hallucination in our models.

## 3 PRELIMINARIES

We first review the existing encoder-decoder framework (Sutskever et al., 2014; Bahdanau et al., 2014) which this work is based on. Let $\boldsymbol{x} = x_1 x_2 \ldots x_S$, be the source input of length $S$ and $\boldsymbol{y} = y_1 y_2 \ldots y_T$ be the target sequence of length $T$. Each token $x_i, y_i$ takes one value from a vocabulary $V$. Our goal is to model the conditional distribution $P(\boldsymbol{y}|\boldsymbol{x}) = \prod_{t=1}^{T} P(y_t|\boldsymbol{y}_{<t}, \boldsymbol{x})$,

where $\boldsymbol{y}_{<t} = y_1 \ldots y_{t-1}$ is the prefix of $\boldsymbol{y}$ up to the $(t-1)^{\text{th}}$ token. The source can be encoded by any neural network function **enc**, such as a convolutional neural network (CNN, LeCun et al., 1990), long-short-term memory (LSTM, Hochreiter & Schmidhuber, 1997), or Transformer (Vaswani et al., 2017). Let $\boldsymbol{s}_1, ..., \boldsymbol{s}_S = \textbf{enc}(x_1, ..., x_S)$.

Define $\boldsymbol{e}_x \in \mathbb{R}^d$ as the $d$ dimensional embedding of token $x$. Then, the probability of each target token is computed as:

$$P\left(y_t \mid \boldsymbol{y}_{<t}, \boldsymbol{x}\right) = \frac{\exp\left(\boldsymbol{v}_t^\top \boldsymbol{e}_{y_t}\right)}{\sum_{y \in V}\left(\exp \boldsymbol{v}_t^\top \boldsymbol{e}_y\right)} \tag{1}$$

where the context vector $\boldsymbol{v}_t$ is given by:

$$\boldsymbol{v}_t = \boldsymbol{a}_t + \boldsymbol{h}_t = \sum_{s=1}^{S} \alpha_{s,t} \boldsymbol{s}_s + \boldsymbol{h}_t \tag{2}$$

In this work, we use a Luong-style attention (Luong et al., 2015) for the first term on the right hand side of Eq. 2, as defined in Eq. 3; and the second term is given by an RNN[1] hidden state at position $t$, as defined in Eq. 4 (let $[\cdot]$ denote concatenation):

$$\alpha_{s,t} = \frac{\exp(\boldsymbol{s}_s^\top \boldsymbol{W} \boldsymbol{h}_t)}{\sum_{s'} \exp(\boldsymbol{s}_{s'}^\top \boldsymbol{W} \boldsymbol{h}_t)} \tag{3}$$

$$\boldsymbol{h}_t = \text{RNN}(\boldsymbol{h}_{t-1}, [\boldsymbol{e}_{y_{t-1}}, \boldsymbol{a}_{t-1}]) \tag{4}$$

In case the encoder-decoder is equipped with a copy mechanism, the generation probability is mixed with a probability of copying from the source (Gu et al., 2016; See et al., 2017):

$$\tilde{P}(y_t | \boldsymbol{y}_{<t}, \boldsymbol{x}) = p_t^{\text{gen}} P(y_t | \boldsymbol{y}_{<t}, \boldsymbol{x}) + (1 - p_t^{\text{gen}}) \sum_{s:x_s = y_t} \beta_{s,t} \tag{5}$$

where $p_t^{\text{gen}}$ is the probability of doing generation instead of copying at step $t$, and $\beta_{s,t}$ is an attention weight that the copy mechanism is paying to position $s$ in the source. The sum is taken over all positions $s$ where the word $x_s$ is the same as $y_t$.

## 4 MODEL

Our approach is based on a confidence score at each decoding position, and the idea is originated from an observation on the hallucinations shown in Figure 1: hallucination often occurs when the table lacks some field that usually exists (e.g. *Occupation*), and the baseline model is trying to make it up (more examples are found in Appendix). In such cases, the generation probabilities for the hallucinated tokens can still be high (e.g. $> 0.5$), so it is difficult to tell hallucination from faithful generation. However, if some usual field is missing in the table, one might tell the difference by assessing the attention in the encoder-decoder. Our preliminary investigation supported this idea: we defined an **attention score** $A_t$ using the length of the attention vector relative to the hidden state, as the following (let $\|\cdot\|$ denote the Euclidean norm):

$$A_t := \frac{\|\boldsymbol{a}_t\|}{\|\boldsymbol{a}_t\| + \|\boldsymbol{h}_t\|} \tag{6}$$

and compared the scores between hallucinated and faithful generations. We found that $A_t$ drops in the hallucinated cases. Then, we refined our attention score with the copy mechanism:

$$\tilde{A}_t := p_t^{\text{gen}} A_t + (1 - p_t^{\text{gen}}) \tag{7}$$

As illustrated in Figure 2, $\tilde{A}_t$ is usually high ($\sim 0.9$) when the model is copying from the source (e.g. *Campbell*), medium ($\sim 0.4$) when it is generating based on some source information (e.g. *<unk>*), and low ($\sim 0.2$) when generating function words, template elements, or when the corresponding field is missing (more examples are found in Appendix).

---

[1]While it is possible our approach could extend to other types of decoders, our current formulation of the confidence score specifically uses RNN with attention.

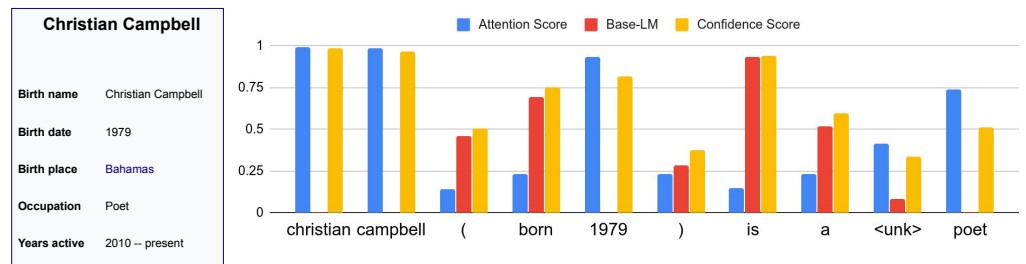

Figure 2: Example of learned attention score, base-LM probability, and confidence score. For content words the base-LM probability is lower, and the confidence score depends more on the attention score.

Thus, the attention score can recognize faithful content words, but it cannot distinguish hallucination from function words and template elements. Therefore, we further introduce an unconditioned language model called **base-LM**. The base-LM does not have access to the source data, so if it can predict a token as precisely as the encoder-decoder, that token is probably an element of a general linguistic construction that does not convey source information. Then, we define **confidence score** as an interpolation between the encoder-decoder $P(y_t|\boldsymbol{y}_{<t}, \boldsymbol{x})$ and the base-LM $P(y_t|\boldsymbol{y}_{<t})$:

$$C\big(y_t \mid \boldsymbol{y}_{<t}, \boldsymbol{x}\big) := \tilde{A}_t P\big(y_t \mid \boldsymbol{y}_{<t}, \boldsymbol{x}\big) + (1 - \tilde{A}_t) P\big(y_t \mid \boldsymbol{y}_{<t}\big) \tag{8}$$

For function words and template elements, we expect both $P(y_t|\boldsymbol{y}_{<t}, \boldsymbol{x})$ and $P(y_t|\boldsymbol{y}_{<t})$ to be high, so the confidence score $C(y_t|\boldsymbol{y}_{<t}, \boldsymbol{x})$ will be high no matter what the attention score is. On the other hand, we expect $P(y_t|\boldsymbol{y}_{<t}, \boldsymbol{x})$ to be higher than $P(y_t|\boldsymbol{y}_{<t})$ for content words, and the confidence score will largely depend on the attention score in this case.

From a high level, our confidence score keeps a balance between two factors:

- How much the model *should* rely on the source for predicting this position, is reflected by combining the encoder-decoder with the base-LM.
- How much the model *actually* relies on the source for this position, is measured by the attention score.

Intuitively, it is reasonable for a system to depend mostly on language modeling to predict function words that make the generation fluent, but it should consult more of the source data to predict content words. For example, given a partial generation "*Christian Campbell is __*", one could predict that the next token is mostly likely "*a*", "*an*" or "*the*", based on language modeling. However, if a model predicts "*American*" as the next token to "*Christian Campbell is an __*", it should be based on a field such as "`Nationality: U.S.`" in the source, rather than the language tendency that "*American*" is likely to appear after the phrase "*is an*". A typical neural network can make predictions based on both reasons with little controllability; this, we contend, is a general cause of hallucination. By introducing the confidence score and enforcing a "confident prior" on our model, we can gain some control on these two.

In order to further strengthen control, we tweak the input-feeding of our RNN models as below:

$$\boldsymbol{g}_t = \text{RNN}\Big(\boldsymbol{g}_{t-1}, \text{SG}\big((1 - \tilde{A}_{t-1})P(y_{t-1}|\boldsymbol{y}_{<t-1})\big)\boldsymbol{e}_{y_{t-1}}\Big) \tag{9}$$

$$\boldsymbol{h}_t = \text{RNN}(\boldsymbol{h}_{t-1}, [\boldsymbol{e}_{y_{t-1}}, \text{SG}(\boldsymbol{a}_{t-1})]) \tag{10}$$

Here, $\boldsymbol{g}_t$ is the hidden-state of the base-LM, and the input embedding $\boldsymbol{e}_{y_{t-1}}$ in Eq. 9 is weighted by a component of the previous confidence score. We found this weighting scheme to decrease dependence of the base-LM on content words, seemingly resulting in a model of soft templates, as shown in Figure 2 (see Appendix for more examples). We also added stop-gradients (SG) to prevent information at the current step from being propagated to previous steps, and used Eq. 10 instead of Eq. 4. They seem to result in more reasonable attention scores.

The confidence score is leveraged in two ways, as we describe in the following:

- At test time, we augment the generation probability with the confidence score, using a calibration technique (Section 4.1). It allows us to weigh more on the confidence of generation, without sacrificing perplexity (Braverman et al., 2019).

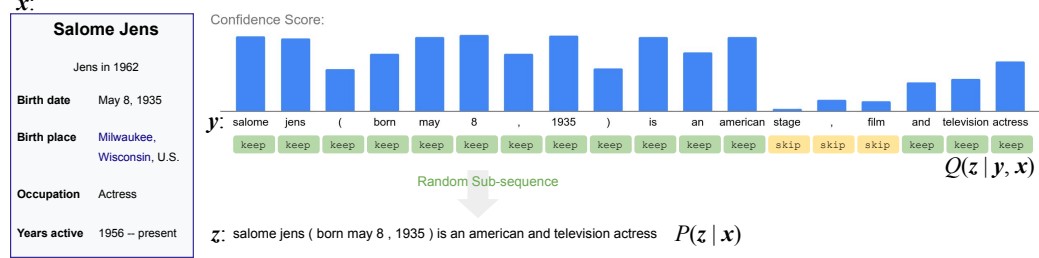

Figure 3: Example of sampling a sub-sequence according to the confidence score. Our variational Bayes objective combines the sampling probability $Q(\boldsymbol{z}|\boldsymbol{y}, \boldsymbol{x})$ and the generation probability $P(\boldsymbol{z}|\boldsymbol{x})$.

- In training, we allow the model to skip some tokens with low confidence score in order to avoid training on noisy references or scarce examples (Section 4.2). However, since the confidence score itself needs to be learned during training, we use a variational Bayes objective to formalize this symbiotic cycle (Kingma & Welling, 2014).

## 4.1 CALIBRATION

In order to use the confidence score to promote faithful generation, we apply a calibration technique (Braverman et al., 2019) which augments the generation probability as follows:

$$\hat{P}^{\kappa}(y_t|\boldsymbol{y}_{<t}, \boldsymbol{x}) := \frac{\mathrm{SG}(C(y_t|\boldsymbol{y}_{<t}, \boldsymbol{x}))^{\kappa}\mathrm{SG}(P(y_t|\boldsymbol{y}_{<t}, \boldsymbol{x}))}{\sum_{w \in V} \mathrm{SG}(C(w|\boldsymbol{y}_{<t}, \boldsymbol{x}))^{\kappa}\mathrm{SG}(P(w|\boldsymbol{y}_{<t}, \boldsymbol{x}))}. \tag{11}$$

Here, $\hat{P}^{\kappa}(y_t|\boldsymbol{y}_{<t}, \boldsymbol{x})$ is a one-parameter family of probability distributions called the calibration of $P(y_t|\boldsymbol{y}_{<t}, \boldsymbol{x})$ to $C(y_t|\boldsymbol{y}_{<t}, \boldsymbol{x})$. Note that SG stops gradients so that only the parameter $\kappa$ is updated during training. In order to learn $\kappa$, we minimize the negative log-likelihood of $\hat{P}^{\kappa}(y_t|\boldsymbol{y}_{<t}, \boldsymbol{x})$ jointly with $P(y_t|\boldsymbol{y}_{<t}, \boldsymbol{x})$ and $P(y_t|\boldsymbol{y}_{<t})$:

$$\mathcal{L}_{\mathrm{joint}}(\boldsymbol{y}, \boldsymbol{x}) := \sum_{t=1}^{T} -\log \hat{P}^{\kappa}(y_t|\boldsymbol{y}_{<t}, \boldsymbol{x}) - \log P(y_t|\boldsymbol{y}_{<t}, \boldsymbol{x}) - \log P(y_t|\boldsymbol{y}_{<t}). \tag{12}$$

Since $P(y_t|\boldsymbol{y}_{<t}, \boldsymbol{x})$ is a special case of $\hat{P}^{\kappa}(y_t|\boldsymbol{y}_{<t}, \boldsymbol{x})$ (namely at $\kappa = 0$), the training perplexity of $\hat{P}^{\kappa}(y_t|\boldsymbol{y}_{<t}, \boldsymbol{x})$ is at most $P(y_t|\boldsymbol{y}_{<t}, \boldsymbol{x})$. In practice, $\kappa$ is initialized as 0 and found converging to a positive value (Section. 5.4). Therefore, the calibration trick can improve confidence of generation without sacrificing perplexity.

## 4.2 TRAINING WITH A VARIATIONAL BAYES OBJECTIVE

In practice, the training data of a conditional text generation task almost always contain noises and/or scarce examples. In order to reduce the impact of such outliers and train a confident generation model, we allow the model to skip some tokens in the training data when it feels unconfident. For this purpose, we use the confidence score to sample a sub-sequence of each training target, and minimize the negative log-likelihood on that confident sub-sequence. However, since the confidence score itself needs to be trained, we use a variational Bayes objective to formalize the problem.

Specifically, for each target $\boldsymbol{y} = y_1 y_2 \ldots y_T$, we define $\boldsymbol{z} = z_1 z_2 \ldots z_R = y_{\iota(1)} y_{\iota(2)} \ldots y_{\iota(R)}$ as a latent sub-sequence of $\boldsymbol{y}$, which consists of confident tokens of length $R$. Here, $\iota : |R| \to |T|$ is an inclusion of indices. We assume that $\boldsymbol{z}$ is generated by the calibrated probability $\hat{P}^{\kappa}(z_r \mid \boldsymbol{z}_{<r}, \boldsymbol{x})$:

$$P(\boldsymbol{z} \mid \boldsymbol{x}) = \prod_{r=1}^{R} \hat{P}^{\kappa}(z_r \mid \boldsymbol{z}_{<r}, \boldsymbol{x}). \tag{13}$$

Then, we connect $P(\boldsymbol{z}|\boldsymbol{x})$ to the probability of training target $P(\boldsymbol{y}|\boldsymbol{x})$. From Bayes rule:

$$P(\boldsymbol{y} \mid \boldsymbol{x}) = \frac{P(\boldsymbol{y}|\boldsymbol{z}, \boldsymbol{x}) \, P(\boldsymbol{z}|\boldsymbol{x})}{P(\boldsymbol{z}|\boldsymbol{y}, \boldsymbol{x})}. \tag{14}$$

We assume that $P(\boldsymbol{y}|\boldsymbol{z}, \boldsymbol{x}) = 1$ for every training example because the training data is uniquely given. Then, we regard $\boldsymbol{z}$ as a sequential "keep/skip" labeling over $\boldsymbol{y}$, and define a probability distribution

$$Q_t := \begin{cases} Q_t(\texttt{keep}) \propto C(y_t | \boldsymbol{z}_{\iota(s)<t}, \boldsymbol{x})^\rho & \text{Where keep means } y_t \text{ is in } \boldsymbol{z} \\ Q_t(\texttt{skip}) \propto \gamma^\rho & \text{Where skip means } y_t \text{ is not in } \boldsymbol{z} \end{cases} \tag{15}$$

to sample a sub-sequence according to the confidence score (Figure 3). Here, $\rho$ and $\gamma$ are hyper-parameters. Now let

$$Q(\boldsymbol{z} \mid \boldsymbol{y}, \boldsymbol{x}) = \prod_{t=1}^{T} Q_t, \tag{16}$$

and the idea is to use $Q(\boldsymbol{z}|\boldsymbol{y}, \boldsymbol{x})$ as an approximation to the unknown posterior $P(\boldsymbol{z}|\boldsymbol{y}, \boldsymbol{x})$ in Eq. 14. By taking $-\log(\cdot)$ of both sides in Eq. 14 and trivially introducing $\log Q(\boldsymbol{z}|\boldsymbol{y}, \boldsymbol{x})$, we get

$$-\log P(\boldsymbol{y} \mid \boldsymbol{x}) = -\log \frac{Q(\boldsymbol{z}|\boldsymbol{y}, \boldsymbol{x})}{P(\boldsymbol{z}|\boldsymbol{y}, \boldsymbol{x})} + \log Q(\boldsymbol{z} \mid \boldsymbol{y}, \boldsymbol{x}) - \log P(\boldsymbol{z} \mid \boldsymbol{x}). \tag{17}$$

Then, we take the expectation $\mathbb{E}_{Q(\boldsymbol{z}|\boldsymbol{y},\boldsymbol{x})}[\cdot]$ of both sides and note that $\mathbb{E}_{Q(\boldsymbol{z}|\boldsymbol{y},\boldsymbol{x})}[\log \frac{Q(\boldsymbol{z}|\boldsymbol{y},\boldsymbol{x})}{P(\boldsymbol{z}|\boldsymbol{y},\boldsymbol{x})}] = KL[Q(\boldsymbol{z}|\boldsymbol{y},\boldsymbol{x})||P(\boldsymbol{z}|\boldsymbol{y},\boldsymbol{x})] \geq 0$, so

$$-\log P(\boldsymbol{y} \mid \boldsymbol{x}) \leq \mathbb{E}_{Q(\boldsymbol{z}|\boldsymbol{y},\boldsymbol{x})}\Big[\log Q(\boldsymbol{z} \mid \boldsymbol{y}, \boldsymbol{x}) - \log P(\boldsymbol{z} \mid \boldsymbol{x})\Big]. \tag{18}$$

The variational Bayes objective is to minimize the upper bound on the right hand side of Eq. 18. In practice, it is computationally expensive to explicitly calculate $\mathbb{E}_{Q(\boldsymbol{z}|\boldsymbol{y},\boldsymbol{x})}[\cdot]$ by enumerating all sub-sequences of $\boldsymbol{y}$, because the number of sub-sequences is exponential to the length $T$. Thus, we apply a Monte Carlo method which calculates $\mathbb{E}_{Q(\boldsymbol{z}|\boldsymbol{y},\boldsymbol{x})}[\cdot]$ by sampling from $Q(\boldsymbol{z}|\boldsymbol{y},\boldsymbol{x})$. In order to back-propagate gradients through the expectation $\mathbb{E}_{Q(\boldsymbol{z}|\boldsymbol{y},\boldsymbol{x})}[\cdot]$ as well, the loss function is given as follows (Paisley et al., 2012):

$$\mathcal{L}_{\text{var}}(\boldsymbol{y}, \boldsymbol{x}) := \frac{1}{K} \sum_{\substack{k=1 \\ \boldsymbol{z}_k \sim Q(\boldsymbol{z}|\boldsymbol{y},\boldsymbol{x})}}^{K} \log Q(\boldsymbol{z}_k \mid \boldsymbol{y}, \boldsymbol{x}) - \log P(\boldsymbol{z}_k \mid \boldsymbol{x})$$
$$+ \lambda \, \text{SG}\Big(\log Q(\boldsymbol{z}_k \mid \boldsymbol{y}, \boldsymbol{x}) - \log P(\boldsymbol{z}_k \mid \boldsymbol{x})\Big) \log Q(\boldsymbol{z}_k \mid \boldsymbol{y}, \boldsymbol{x}). \tag{19}$$

Here, $K$ is the number of samples taken, and $\lambda$ is a hyper-parameter controlling how fast the gradients go through $\mathbb{E}_{Q(\boldsymbol{z}|\boldsymbol{y},\boldsymbol{x})}[\cdot]$. However, since we define $P(\boldsymbol{z}|\boldsymbol{x})$ in Eq. 13 by the calibrated probability, which only learns one parameter $\kappa$, we add joint learning terms into Eq. 19 to make the final objective:

$$\mathcal{L}(\boldsymbol{y}, \boldsymbol{x}) := \frac{1}{K} \sum_{\substack{k=1 \\ \boldsymbol{z}_k \sim Q(\boldsymbol{z}|\boldsymbol{y},\boldsymbol{x})}}^{K} \log Q(\boldsymbol{z}_k \mid \boldsymbol{y}, \boldsymbol{x}) + \mathcal{L}_{\text{joint}}(\boldsymbol{z}_k, \boldsymbol{x})$$
$$+ \lambda \, \text{SG}\Big(\log Q(\boldsymbol{z}_k \mid \boldsymbol{y}, \boldsymbol{x}) - \log P(\boldsymbol{z}_k \mid \boldsymbol{x})\Big) \log Q(\boldsymbol{z}_k \mid \boldsymbol{y}, \boldsymbol{x}). \tag{20}$$

In order to sample sub-sequences from $Q(\boldsymbol{z}|\boldsymbol{y},\boldsymbol{x})$, we apply the same Gumbel-max trick as in Kool et al. (2019). Although the random sampling is purely based on a learned probability distribution, without any constraints to make it fluent, surprisingly the model still learns to generate fluent text.

## 5 EXPERIMENTS

Although our approach could apply to many conditional text generation tasks, in this work we consider data-to-text generation, in which the source is some structured data and the target is natural language text describing the data. Usually, the data have concise semantics and simple structure, which makes it easy to check the facticity of the generation.

| Model | | Automatic Metrics | | | | Human evaluation | |
| | BLEU | PARENT (Prec. / Rec. / F1) | Avg Len. | | Faithful % | Avg Cov. | Fluency % |
|---|---|---|---|---|---|---|---|
| BERT-to-BERT (Rothe et al., 2019) | 44.83 | 77.62 / 43.00 / 53.13 | 20.9 | | 77.6 | 4.33 | **98.5** / 99.4 |
| Structure-Aware Seq2Seq (Liu et al., 2018) | **45.36** | 73.98 / **44.02** / 52.81 | 23.1 | | 66.1 | **4.47** | 88.6 / 99.7 |
| Pointer-Generator (See et al., 2017) | 41.07 | 77.59 / 42.12 / 52.10 | 19.1 | | 80.3 | 4.24 | 93.1 / 96.0 |
| Confident BERT-to-RNN (This Work) | 33.30 | 77.98 / 37.21 / 47.90 | 16.6 | | 85.2* | 3.90 | 92.3 / 94.1 |
| Confident Pointer-Generator (This Work) | 38.10 | **79.52** / 40.60 / 51.38 | 17.0 | | 86.8* | 4.05 | 95.4 / 96.3 |
| +threshold=0.125 | 36.62 | **80.15** / 39.59 / 50.50 | 16.4 | | **90.7*** | 4.01 | 91.6 / 92.2 |

Table 1: Performance on WikiBio test set. Two Fluency measures differ in whether to include sentences graded as *Mostly Fluent*. Starred numbers are statistically significant against baselines ($p < .001$), by bootstrap test.

## 5.1 DATASET AND EVALUATION METRICS

The WikiBio dataset (Lebret et al., 2016) contains 728,321 biographies paired with infoboxes, taken from the Sep-2015 dump of English Wikipedia, and splitted into train/valid/test sets in a $8 : 1 : 1$ ratio. The biography text is the first sentence of the Wikipedia page (26.1 words on average). Infoboxes have 12.1 non-empty fields on average.

For automatic metrics, we report BLEU (Papineni et al., 2002), as well as PARENT (Dhingra et al., 2019), a metric designed to mitigate the shortcomings of BLEU on structured data-to-text generation.

For human evaluation, we obtain crowd-source annotations on examples randomly chosen from predictions on the WikiBio test set, the same 1000 for each model. Examples from different models are mixed and randomly shuffled, with model names hidden from the raters. We instruct the raters to grade on each of 3 criteria: faithfulness, coverage, and fluency, as below:

- Faithfulness (precision) - We define a sentence to be faithful if all the information in the proposed sentence is supported by the table or the reference. A single hallucinated piece of information makes the sentence non-faithful.
- Coverage (recall) - The number of table cells that contain information present in the sentence.
- Fluency - A sentence is defined to be fluent if it is clear, natural, and grammatically correct. Raters choose among three options: *Fluent*, *Mostly Fluent*, *Not Fluent*.

An ideal system would always produce fluent and faithful text with high coverage. The complete annotation instruction is given in the Appendix. We assigned 5 raters who are well-trained on this task, and conducted 5-way annotation on a pilot batch of 100 examples. The unanimous agreement rate is 86% for faithfulness and 99% for fluency, with 74% of the examples judged as faithful and 98% as fluent.

## 5.2 EXPERIMENT SETTING

We compare the following systems:

- BERT-to-BERT (Rothe et al., 2019): A Transformer encoder-decoder model (Vaswani et al., 2017) where the encoder and decoder are both initialized with BERT (Devlin et al., 2018).
- Structure Aware Seq2Seq (Liu et al., 2018): A state-of-the-art method on the WikiBio dataset in terms of BLEU.
- Pointer-Generator (See et al., 2017): A Seq2Seq with attention and copy mechanism (our implementation).
- Confident BERT-to-RNN (This Work): A Transformer encoder initialized with BERT checkpoint, and a GRU (Cho et al., 2014) decoder with our confident decoding method.
- Confident Pointer-Generator (This Work): Pointer-Generator model with confident decoding.

We built our systems using Tensorflow (Abadi et al., 2016). Infoboxes are linearized into sequences, with field names and values separated by special tokens. For BERT-to-BERT and Confident BERT-to-RNN, we pre-trained a BERT checkpoint on the Books corpus (Zhu et al., 2015) only, since the original BERT was trained on Wikipedia that overlaps with the test targets in WikiBio. Other hyper-parameter settings are given in Appendix.

|  | BLEU | PARENT (Prec. / Rec. / F1) | Avg Len. |
|---|---|---|---|
| Confident Pointer-Generator (This Work) | 38.10 | **79.52** / 40.60 / 51.38 | 17.0 |
| No base-LM | 39.39 | 78.77 / 41.55 / 52.08 | 17.9 |
| No calibration | 37.89 | 79.47 / 40.47 / 51.26 | 16.9 |
| No variational | **41.29** | 78.25 / **42.40** / **52.52** | 18.9 |
| Pointer-Generator | 41.07 | 77.59 / 42.12 / 52.10 | 19.1 |
| Truncated | 35.50 | 77.68 / 38.16 / 48.66 | 17.1 |

Table 2: Ablative tests on three components of our confident decoding method, and a truncation test.

## 5.3 MAIN RESULTS

Table 1 shows the results. According to human evaluation, our approach gives a clear improvement in faithfulness over the baselines, with some drop in coverage. To further measure the validity of our confidence score, we postprocessed the output to remove words with lower confidence than $0.125$. This thresholding technique gives further gains to faithfulness, while sacrificing some fluency.

Among the automatic metrics, PARENT precision and recall seem correlated to faithfulness and coverage respectively, and our approach achieves the highest precision score. BLEU, perhaps because of its length penalty that rewards longer generations, seems more correlated to coverage rather than faithfulness. Regarding the baselines, we see that BERT-to-BERT is the most fluent while Pointer Generator is the most faithful, suggesting that pretraining might help fluency while the copy mechanism can be valuable for faithfulness.

## 5.4 ABLATIONS

Our confident decoding method has three novel components: **(1)** The use of a base-LM to define confidence score; **(2)** The calibration technique to adjust output probability; and **(3)** The variational Bayes objective to train a confident model. In this section, we assess the effects of each component by an ablative study. We start from the Confident Pointer-Generator, and in each test replace one component by a trivial alternative: **(1)** In order to assess the effects of using the base-LM in the confidence score, we instead use $P(y_t|\boldsymbol{y}_{<t}, \boldsymbol{x})$ directly as confidence, and train models with the same hyper-parameter search. The results on WikiBio are shown in Table 2 as "No base-LM". **(2)** We use $P(y_t|\boldsymbol{y}_{<t}, \boldsymbol{x})$ instead of $\hat{P}^\kappa(y_t|\boldsymbol{y}_{<t}, \boldsymbol{x})$ at test time ('No calibration"), to assess the effects of calibration. The model is the same as Confident Pointer-Generator. The learned $\kappa$ was $0.035$. **(3)** Instead of the variational Bayes objective, we use the joint training loss $\mathcal{L}_{\text{joint}}$ in Eq. 12 without sampling sub-sequences from training targets (No variational).

As we can see from Table 2, all three components improve PARENT precision. While the improvement by calibration is the smallest, the technique also improves PARENT recall and BLEU score at the same time, making it an easy choice. The other techniques trade recall for precision, making them useful for tasks that require a high degree of faithfulness. When all three components are disabled, the model is exactly the same as our implementation of the Pointer-Generator. Every component improves PARENT precision upon it as well. Especially, comparing Pointer-Generator with "No variational" shows again that joint training with calibration improves all metrics.

We also note that the average lengths of generations by our confident decoding models are shorter. While there exists heuristics such as length penalty (Wu et al., 2016) to encourage longer generation at inference time, shorter generation is not trivial. In the "Truncated" setting, we truncate predictions by the Pointer-Generator two words each to match the average length of Confident Pointer-Generator. The PARENT precision by our confident decoding method is not trivially achieved by truncation.

## 6 CONCLUSION

In this work, we have proposed a confidence oriented decoder that achieved more faithful generation on the WikiBio dataset than existing state-of-the-art approaches. Our method is general in principle, so it could potentially be combined with other approaches and/or adapted to other forms of conditional text generation such as document summarization and machine translation.

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

## 7 APPENDIX

### 7.1 HYPER-PARAMETERS

| Model | Warmup Steps | Learning Rate | RNN Dropout | RNN Dim. | Beam Size | $\rho$ | $\gamma$ | $K$ | $\lambda$ |
|---|---|---|---|---|---|---|---|---|---|
| Confident BERT-to-RNN | 40000 | 0.05 | 0.1 | 768 | 8 | 0.75 | 1/8 | 4 | 1/64 |
| Confident Pointer-Generator | None | 0.0005 | 0.2 | 200 | 8 | 0.5 | 1/16 | 4 | 1/4 |

Table 3: Hyper-parameters. We did a hyper-parameter search for $\rho$, $\gamma$ and $\lambda$, within the range $[0.25, 0.5, 0.75, 1]$, $[1/16, 1/8, 1/4]$ and $[1/64, 1/16, 1/4, 1]$, respectively. Model selection is based on PARENT F1.

For Pointer-Generator and Confident Pointer-Generator, we use GloVe (Pennington et al., 2014) as the input word embedding and truncate the vocabulary size to 5,000; the two models share the same hyper-parameter settings. For our confident decoding models, there are additional hyper-parameters $\rho, \gamma$ as defined in Eq. 15, and $K, \lambda$ as defined in Eq. 19. The hyper-parameters are shown in Table 3. The optimizer is Adam (Kingma & Ba, 2015).

### 7.2 HUMAN EVALUATION INSTRUCTIONS

Here are the detailed instructions for our human evaluation template:

*A writer has the following background knowledge and is given a table:*
<Reference sentence shown as background knowledge>
<Table>

*The writer read the table and produced the following sentence:*
<Sentence generated by model>
*We wish to evaluate the quality of the sentence.*

*1. How **fluent** is the sentence?*

- *Fluent: It is clear, natural, and the grammar is correct. It reads like if it was found in a book.*
- *Mostly Fluent: It has a few errors or it does not sound natural, but you can understand it.*
- *Not Fluent: It has many errors and/or you can hardly understand it.*

*Examples:*

| Sentence | Decision |
|---|---|
| *alfred angas scott ( 1875 - 1923 ) was a motorcycle designer born in manningham bradford , who lived in* | ***Not fluent****: the sentence stops abruptly. It is unnatural, it does not make any sense.* |
| *alfred angas scott ( 1875 - 1923 ) was a motorcycle designer designer born in manningham bradford .* | ***Mostly fluent****: the repetition "designer designer" is not natural, but the sentence makes sense.* |
| *alfred angas scott ( 1875 - 1923 ) was a motorcycle designer , was born in manningham bradford , lived in england u.k. , was british by nationality , and was buried at the undercliffe cemetery in bradford .* | ***Mostly fluent****: there are no mistakes, but the sentence contains too much information. This is unnatural.* |

*2. Please compare carefully the content of the sentence to the content of the table. How many cells from the table did the writer use to produce the sentence? (Click on the cells in the table above to update the counter)*

*3. A sentence is **faithful** if it contains only information supported by the table or the writer's background knowledge. It should not add any additional information, even if the information is true or interesting. Please compare once again the content of the sentence to the content of the table and background knowledge. How faithful is the sentence?*

- *Faithful: every part of the sentence is supported by the table and/or background knowledge.*
- *Mostly Faithful: every part of the sentence can be linked to some evidence in the table or the background knowledge, but it is not fully supported. This should only be used for rare edge cases.*
- *Not Faithful: The sentence contains information that is not supported by the table or background knowledge.*

*The examples are based on the following background knowledge and table:*

*alfred angas scott ( 1875 - 1923 ) was a british motorcycle designer .*

| birth date | birth place | death date | nationality | residence | occupation |
|---|---|---|---|---|---|
| 1875 | manningham, bradford, united kingdom | 1923 | british | england, uk. | motorcycle designer and manufacturer |

*Examples:*

| Sentence | Decision |
|---|---|
| alfred angas scott ( 1875 - 1923 ) was a british motorcycle designer and founder of the scott motorcycle company . | **Coverage**: 4 
 **Not faithful**: the table does not mention that A. A. Scott was the founder of the Scott Company . |
| alfred angas scott ( 1875 - 1923 ) was a european motorcycle designer . | **Coverage**: 4 
 **Faithful**: All the information is supported because England is located in Europe. |

We have also discussed with the lead annotators about many other examples. We have made sure that: (a) Valid inferences (e.g. inferring nationality from birth place) are considered faithful; (b) If a piece of information exists or can be inferred from the table, the corresponding cell should be highlighted, even if the information was also in the background knowledge; (c) Only one cell should be highlighted for one piece of information.

| Table | Pointer Generator | Confident Pointer Generator |
|---|---|---|
| Team missing | rohan robertson ( born 21 august 1961 ) is a former australian rules footballer who played with carlton in the victorian football league . | rohan robertson ( born 21 august 1961 ) is a former australian rules footballer who played in the victorian football league . |
| Middle name missing | walter herbert smallwood ( april 24 , 1893 – april 29 , 1967 ) was a pitcher in major league baseball . | walter smallwood ( april 24 , 1893 – april 29 , 1967 ) was a pitcher in major league baseball . |
| Residence missing | nellie wong ( born september 12 , 1934 in oakland , california ) is an american poet , activist , feminist , and feminist activist who lives and works in los angeles , california , united states , where she | nellie wong ( born september 12 , 1934 ) is an american poet and activist . |
| Name missing | gene ( born december 21 , 1941 ) is a former american football tight end in the national football league . | ( born december 21 , 1941 ) is a former american football tight end in the national football league for the minnesota vikings . |
| Nationality & occupation missing | constant vanden stock ( june 13 , 1914 – april 19 , 2008 ) was an american figure skater . | constant vanden stock ( june 13 , 1914 – april 19 , 2008 ) was a . |
| Table fields are: name, nationality, years and teams | richard lloyd lloyd is a british racing driver who won the gti engineering championship in 1982 , driving with gti engineering , richard lloyd racing of gti engineering , and lloyd lloyd racing at the age of 14 . | richard lloyd is a former racing driver . |
| District: 3rd | robert i. marshall ( born october 16 , 1946 in wilmington , delaware ) is an american politician and a democratic member of the delaware senate since january 9 , 1979 representing district 41 . | robert i. marshall ( born october 16 , 1946 in wilmington , delaware ) is an american politician and a democratic member of the delaware senate since january 9 , 1979 . |
| Known for: English and Welsh dictionary | thomas edwards ( 1779 – 4 june 1858 ) was an english author , <unk>, and <unk>, who spent most of his life in the english and english literature of the english and english dictionary literature . | thomas edwards ( 1779 – 4 june 1858 ) was a welsh author . |

Table 4: Generation examples. Red tokens are hallucinated.

### 7.3 HALLUCINATION EXAMPLES

In Table 4, we show some typical case where the Pointer Generator baseline hallucinates but the Confident Pointer Generator does not. The examples are taken from the WikiBio validation set.

In the first five examples (i.e. "*Rohan Robertson*", "*Walter Smallwood*", "*Nellie Wong*", "*tight end*" and "*Constant Vanden Stock*"), some information is missing in the table while the baseline made it up. Our Confident Pointer Generator model learns to omit the missing fields, although by doing this, two of the generated sentences are no longer fluent (i.e. "*tight end*" and "*Constant Vanden Stock*").

In the next example (i.e. "*Richard Lloyd*"), the baseline seems to have learned weird language modeling from some similar training points, and tries to generate more than the table contents; our Confident Pointer Generator is more conservative.

In the last two examples (i.e. "*Robert I. Marshall*" and "*Thomas Edwards*"), there are corresponding fields in the table but the baseline didn't learn to generate the text correctly, possibly because these

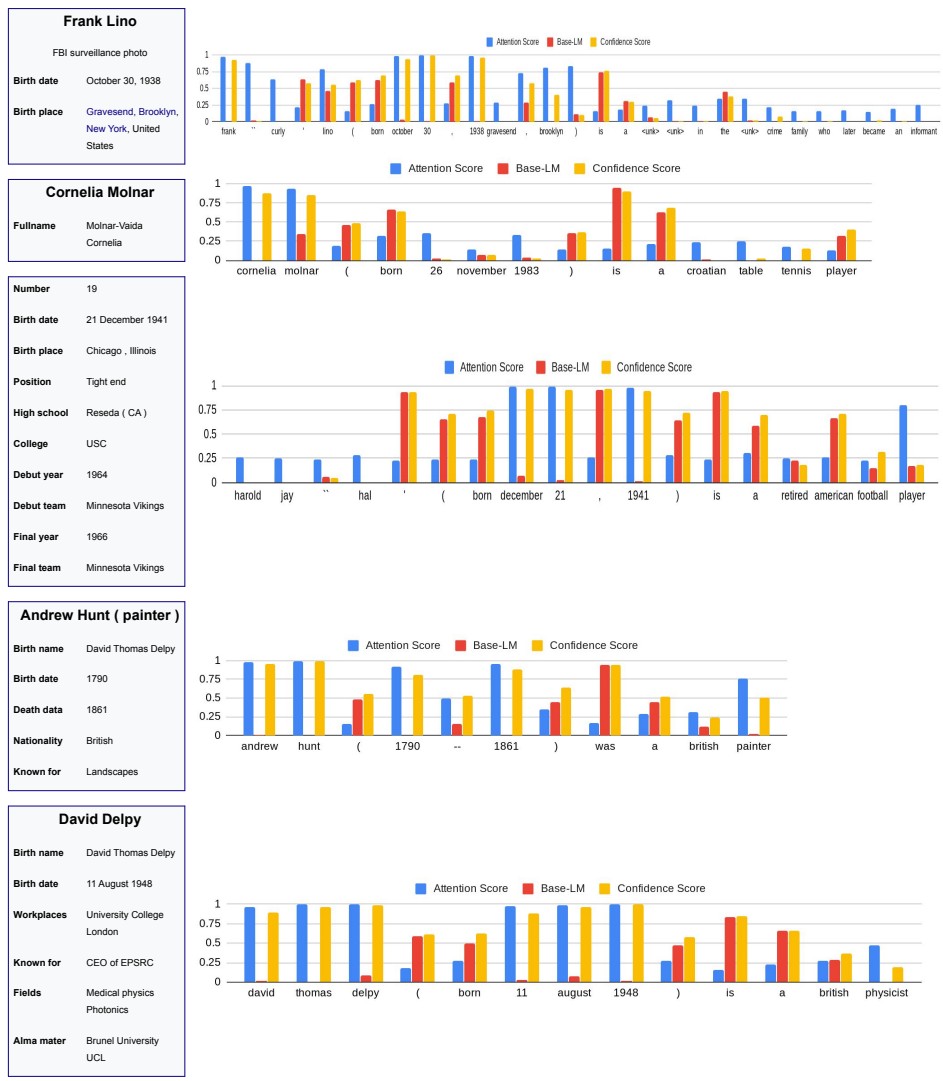

Figure 4: Examples of the learned attention score, base-LM probability and confidence score.

fields should not be simply copied but there are not enough training data to learn the correct generation. Our Confident Pointer Generator model simply omits these fields.

### 7.4 ATTENTION SCORE, BASE-LM PROBABILITY AND CONFIDENCE

In Figure 4, we show the attention score, base-LM probability and confidence score learned by our Confident Pointer-Generator model. The examples are taken from WikiBio validation set.

The first three examples (i.e. "*Frank Lino*", "*Cornelia Molnar*" and "*tight end*") are reference sentences, and our model successfully detected the tokens not supported by the table (i.e. *Occupation* is missing in the *Frank Lino* table, the *Cornelia Molnar* table only has a name, and the *tight end* table does not have a name), which suggests that our model may correctly skip unsupported tokens during training.

The last two examples (i.e. "*Andrew Hunt*" and "*David Delpy*") are model generated sentences, and we can see that the attention score intuitively measures how related a token is to the source table, and the base-LM successfully learned soft templates.

