# OpenReview forum: "Sticking to the Facts: Confident Decoding for Faithful Data-to-Text Generation"
_ICLR.cc/2020/Conference — Reject_

### Official Review · AnonReviewer2 · 2019-10-23
**Official Blind Review #2**

**Rating:** 6

**Review:**

This paper studies the problem of data-to-text generation so that the generated text stays truthful to the data source. The idea of the paper is use a learned confidence score as to how much the the encoder-decoder is paying attention to the source. The paper includes several components, 1. unconditioned language model to incorporate the confidence score, 2. use calibration techniques to adjust the output probability; 3. variational bayes objective to learn the confidence score.

The paper has good motivations and is quite well-written. The problem is of great pragmatic interest. In the experimental part, the authors demonstrate the effectiveness of the proposed algorithm.

1. For training part, regarding the language model and  variational bayes objective being trained jointly, does it have convergence problem? What is the motivation of not training them jointly?
2. Will the code be released and the human evaluation be published?
3. There are some importance baseline missing, such as [1], [2], [3]

[1] Marcheggiani, Diego, and Laura Perez-Beltrachini. "Deep graph convolutional encoders for structured data to text generation." arXiv preprint arXiv:1810.09995 (2018).
[2] Ratish Puduppully, Li Dong, and Mirella Lapata. "Data-to-text Generation with Entity Modeling." arXiv preprint arXiv:1906.03221 (2019).
[3] Ma, Shuming, et al. "Key Fact as Pivot: A Two-Stage Model for Low Resource Table-to-Text Generation." arXiv preprint arXiv:1908.03067 (2019).


**Experience Assessment:**

I have read many papers in this area.

**Review Assessment: Checking Correctness Of Derivations And Theory:**

I assessed the sensibility of the derivations and theory.

**Review Assessment: Checking Correctness Of Experiments:**

I carefully checked the experiments.

**Review Assessment: Thoroughness In Paper Reading:**

I read the paper at least twice and used my best judgement in assessing the paper.

---

> ### Author Response · Authors · 2019-11-14
> **Response to Reviewer #2**
>
> Thanks for the informative review. Reviewer #2 suggested some more related works and wanted to know if we will publicly release our code and data.
>
> > Regarding our code and human evaluation data:
>
> Yes, we will release our model predictions and human evaluations soon, and open source the code upon publication of this work. We have also modified our paper and added generation examples and more details about our human evaluation process.
>
> > Regarding related works:
>
> [2] has already been cited by the previous version of our paper, and we have added [1] and [3] to the revised version. Thanks!
>
> The approaches described in [1][2][3] are not directly comparable to our model: [1] and [2] are not tested on the WikiBio dataset, especially [2] has many data specific model designs that may not be easily applied to WikiBio. (However, please see our discussion with Reviewer #3 about results on the RotoWire dataset.) [3] has a different focus on low resource table-to-text generation using only limited training data. Therefore, we did not compare with them in our experiments.
>
> > regarding the language model and  variational bayes objective being trained jointly, does it have convergence problem?
>
> There seems to be no particular convergence problems; all of our models are tested upon convergence. The training process is like, at the beginning all tokens are unconfident, then the entropy term in our variational Bayes loss pushes up the confidence of some tokens; the model learns to generate highly confident tokens first, such as names and birth dates; then, gradually the model learns to generate longer sentences. It is quite sensitive to hyper-parameters how conservative the final model becomes.

---

### Official Review · AnonReviewer1 · 2019-10-24
**Official Blind Review #1**

**Rating:** 3

**Review:**

This paper aims to solve the unfaithful generation problem for a specific data-to-text generation task, i.e. wikibio dataset. The wikibio dataset has a specific feature, where the output doesn't often reflect the input info box. This will cause the traditional seq2seq-style neural generation models to hallucinate frequently since the training objective is often based on word likelihood.

The paper thus design a confidence scorer that estimates whether a word should be generated according to the source information. This score is used in both training and testing. In training, it helps avoid learn to generate the low confidence words. In testing, it is used to adjust output probabilities.

Overall, I think this is an interesting idea. However, the design of confidence score highly rely on the attentions calculates from the generation process, and whether attentions can be reliably estimated is questionable. Maybe it would be useful to show some statistics (not just manually picked examples) on the hallucinated words, and see what's the portion of them are due to "flattened" attentions.

Furthermore, the experimental results are not convincing. The generations of the proposed models are significantly shorter (might be the result of training, see my comment below about 4.3), the results are mixed, both coverage and fluency are worse. Wrt results, since the dataset is from Wiki, BLEU should be pretty indicative of the generation quality. And we do see significant drop of the proposed model.


More comments:
- Eq 6 needs to be better explained. I don't know if this is the common way to calculate attentions, or I misunderstood the equation.

- In 4.3, I'm not sure if I can understand it correctly. When the authors say "minimize the negative log-likelihood on the confidence sub-sequence", does it mean words not in the subsequences are ignored? Won't this hurt the language modeling part? I.e. cause the ungrammaticality? Is this why the fluency scores are low in Table 2?

- If the authors want to show their model improve faithfulness, sample outputs should be shown.

**Experience Assessment:**

I have published in this field for several years.

**Review Assessment: Checking Correctness Of Derivations And Theory:**

I assessed the sensibility of the derivations and theory.

**Review Assessment: Checking Correctness Of Experiments:**

I carefully checked the experiments.

**Review Assessment: Thoroughness In Paper Reading:**

I read the paper thoroughly.

---

> ### Author Response · Authors · 2019-11-14
> **Response to Reviewer #1**
>
> Thanks for the thoughtful review. Reviewer #1 has concerns about our model design and does not fully agree with our evaluation. We address these below.
>
> In our revised paper, we have added generation examples and more details about our human evaluation process to further strengthen our points. We will also release our model predictions and human evaluations soon, and open source the code upon publication of this work.
>
> > whether attentions can be reliably estimated is questionable. Maybe it would be useful to show some statistics (not just manually picked examples) on the hallucinated words, and see what's the portion of them are due to "flattened" attentions.
>
> In our revised paper, we have added more examples showing the attention score. Typically, the attention score is high (~0.9) when the model is copying from source (e.g. “Campbell” in Figure 2), medium (~0.4) when the model is generating based on some information from the source (e.g. “<unk>” in Figure 2), and low (~0.2) when the model is generating template elements (e.g. “is” in Figure 2). As a quantitative evaluation, please recall that we can boost the faithfulness by four points in human evaluation simply by removing low confidence tokens in post-processing: this clearly demonstrates that low confidence score is strongly correlated with hallucinated words.
>
> > the experimental results are not convincing. The generations of the proposed models are significantly shorter
>
> We regard generating shorter sentences as a feature rather than shortcome. The sentence length itself is not an issue here; in this work we focus on faithfulness. We have shown that our novel techniques can improve faithfulness, but sometimes at the cost of slightly reducing coverage. These techniques are general in principle so they can potentially be combined with other techniques, for example the ones that increase coverage, to make better systems.
>
> > the results are mixed, both coverage and fluency are worse
>
> Actually, our Confident Pointer-Generator model has a better fluency than the baseline Pointer-Generator.
>
> > BLEU should be pretty indicative of the generation quality. And we do see significant drop of the proposed model
>
> BLEU is a bad metric for measuring faithfulness, and it is strongly correlated with sentence length. As a matter of fact, one can simply boost the BLEU score of the baseline Pointer-Generator model from 41 to 45 by setting Wu et al. (2016)’s length penalty to 2.0 at inference time; but this will hurt PARENT precision and decrease the faithfulness by 10 points according to our human evaluation. At least for the Wikibio dataset, we should trust PARENT precision more than BLEU score for measuring faithfulness, which we believe is one of the major points established by Dhingra et al. (2019).
>
> > Eq 6 needs to be better explained.
>
> Thanks. We have edited our paper to make the motivation clearer.
>
> > When the authors say "minimize the negative log-likelihood on the confidence sub-sequence", does it mean words not in the subsequences are ignored?
>
> Yes, exactly.
>
> > Won't this hurt the language modeling part? I.e. cause the ungrammaticality? Is this why the fluency scores are low in Table 2?
>
> No. Surprisingly, this won’t hurt fluency and mostly doesn’t cause ungrammaticality. Actually, our Confident Pointer-Generator model has a better fluency than the baseline Pointer-Generator. We will release data soon including the predictions by our model. And we will release our code upon publication of this work.
>
> The fact that it doesn’t hurt fluency is also a surprise to us. The model does seem to generate some ungrammatical sentences when we use greedy search for decoding; but the results become mostly fluent after we adopted beam search with beam size 8.
>
> We have tried more conservative approaches, such as pretraining the base-LM on the training corpus without subsequence sampling. Those methods do not necessarily improve fluency, but hurt faithfulness instead.
>
> > If the authors want to show their model improve faithfulness, sample outputs should be shown.
>
> Thanks. We have added more generation examples and human evaluation details to our revised paper, and welcome the critical judgement. We will also release our human evaluation data soon.

---

### Official Review · AnonReviewer3 · 2019-11-02
**Official Blind Review #3**

**Rating:** 8

**Review:**

This paper presents a method for conditional text generation that has higher factual precision, minimizing hallucination of facts. The method involves predicting confidence of generation at each time step and using this confidence measure to skip tokens during generation and calibrate output probabilities in test time. Their method achieves SoTA performance on automatically measured precision and human evaluated "faithfulness." However their method does see a drop in recall (automatic metric and human evaluation).

Comments and issues,
- The intuitive explanation for the confidence score is a little confusing. In Section 4, page 3, you say that "If a token is likely a content word (i.e. when its generation probability by the encoder-decoder is much higher than the unconditioned language model), but the attention score is low, then the token might not be predicted based on the source, and could be hallucination." However, this doesn't seem like an airtight conclusion. Isn't it possible that the base-LM and enc-dec model have similar probabilities for a content word with the enc-dec attention being low? This seems possible given your observation that low attention to the source is what may be causing content hallucination. This same thing is essentially restated in section 4.1 "we expect P(y_t |y_<t, x) to be higher than P(y_t | y_<t) for content words so the confidence score will largely depend on the attention score", which seems more tangled up since P(y_t |y_<t, x) inherently depends on the attention score. This is all clarified when you explain the alteration made to the base-LM. I would recommend rewording/rearranging some of the earlier explanation for the efficacy of the confidence score since it seems that the alteration to the base-LM is an essential part of the explanation.
- Need some explanation for Equation 6. I don't really get the intuition behind it.
- The presented results are pretty good! However, I would like to see some numbers on average score across a few runs.
- It would also be good to see results on one more dataset like E2E.
- Provide a little more detail on human evaluation, you don't even mention if the evaluation was done with crowd-workers or another pool of people like grad students. How many annotators? What is the inter-annotator agreement? What was the prompt/structure? Human evaluation of models is notoriously difficult, more details would give some more weight to the results.

I think this is a well written paper with thought out experiments. I recommend it be accepted to ICLR. I'd also be curious to see some future work that improves, or at least maintains recall, while keeping the higher precision.

Minor requests/recommendations:
- Include more examples of generations. Could be an appendix.


**Experience Assessment:**

I do not know much about this area.

**Review Assessment: Checking Correctness Of Derivations And Theory:**

I assessed the sensibility of the derivations and theory.

**Review Assessment: Checking Correctness Of Experiments:**

I assessed the sensibility of the experiments.

**Review Assessment: Thoroughness In Paper Reading:**

I read the paper at least twice and used my best judgement in assessing the paper.

---

> ### Author Response · Authors · 2019-11-14
> **Response to Reviewer #3**
>
> Thanks for the detailed review. Reviewer #3 has suggested motivating our model designs better, describing more details about our human evaluation, and adding more generation examples. We have added these in our revised paper.
>
> We have also done some extra work on additional runs and more datasets, as discussed below:
>
> > I would like to see some numbers on average score across a few runs
>
> We do not have an average across multiple runs, but a second run of our model suggests that: similar BLEU and PARENT scores can be achieved by different runs, but the best performing hyper-parameters vary -- the chosen \rho, \gamma and \lambda reported in our paper do not always give the best results; it is better to sweep on these hyper-parameters.
>
> > It would also be good to see results on one more dataset like E2E.
>
> Actually, we had results on a second dataset: the RotoWire (Wiseman et al. 2017). We did not use E2E because E2E seems simpler and has less source-reference divergence; we wanted to test on a more complicated and hallucination-prone dataset. Our results on RotoWire are as follows:
>
> Entity Modelling (Puduppully et al. 2019): BLEU 16.37  PARENT Prec. 34.68 Rec. 36.79 F1 34.47 Avg. Len. 295
> Content Planning (Puduppully et al. 2018): BLEU 16.85 PARENT Prec. 35.40 Rec. 40.41 F1 36.59 Avg. Len. 332
> Pointer-Generator: BLEU 9.15 PARENT Prec. 37.68 Rec. 36.48 F1 35.94 Avg. Len. 251
> Confident Pointer-Generator: BLEU 8.40 PARENT Prec. 42.64 Rec. 35.23 F1 37.69 Avg. Len. 233
>
> It seems that our Confident Pointer-Generator achieves SoTA PARENT Precision on RotoWire as well. However, we did not report these results in our paper because we did not conduct human evaluation.
>
> > I'd also be curious to see some future work that improves, or at least maintains recall, while keeping the higher precision.
>
> Absolutely. To extend this approach and achieve high precision text generation on more complicated datasets is one of the major topics we are working on.

---

> > ### Comment · AnonReviewer3 · 2019-11-14
> > **Replying to author response**
> >
> > > Multiple runs
> > I think this is worth mentioning in the paper. Having to do a new hyperparameter search for each new random restart isn't ideal and useful information for anyone who might want to adopt the model.
> >
> > > RotoWire results
> > In spite the lack of human eval results, I'd recommend including the RotoWire results somewhere in the paper, even if it's in another Appendix. It's a result that reinforces the efficacy of the method and is reassuring to see.
> >
> > > Human eval
> > Thank you for an appendix with details! It would still be important to see inter-annotator agreement since these types of model evaluations are notoriously high variance. Have you considered getting comparative human eval numbers, where you show a worker generations from two models and ask to rank them on fluency and faithfulness? This type of evaluation may be more reliable.
> >
> > Lastly, the added explanations for the attention score and the confidence score are helpful add clarity to the paper.

---

> > > ### Author Response · Authors · 2019-11-14
> > > **Re:**
> > >
> > > Thanks for the comments!
> > >
> > > We will consider further updating our paper to reflect the additional results and discussions here.
> > >
> > > Regarding human eval, just a quick notice that we have reported inter-annotator agreement in Section 5.1: We assigned 5 raters who are well-trained on this task, and conducted 5-way annotation on a pilot batch of 100 examples. The unanimous agreement rate is 86% for faithfulness and 99% for fluency, with 74% of the examples judged as faithful and 98% as fluent.
> > >
> > > As for side-by-side comparisons between models, we have considered it but did not conduct because it will depend on two models. We wanted some absolute measure for each model that can be compared across, without depending on other models or baselines.

---

### Official Review · AnonReviewer4 · 2019-11-03
**Official Blind Review #4**

**Rating:** 3

**Review:**

The authors propose several approaches to making a data-to-text generation system more precise, that is, less prone to hallucination.  In particular, they propose an attention score, which attempts to measure to what degree the model is relying on its attention mechanism in making a prediction. This attention score is used to weight a mixture distribution (a "confidence score") over the generation model's next-word distribution and the next-word distribution of an unconditional language model. The learned conditional distribution can then be calibrated to the confidence score. The authors also propose a variational-inference inspired objective, which attempts to allow the model to ignore certain tokens it isn't confident about. The authors evaluate their approach on the WikiBio dataset, and find that their approaches make their system more precise, at the cost of some coverage.

This paper is well motivated, timely, and it presents several interesting ideas. However, I think parts of the proposed approach need to be better justified. In particular:

-  What justifies defining the attention score A_t in this way? First, is there an argument (empirical or otherwise) for using the magnitude of the attention vector (rather than some other statistic)? Is it obvious that if the attention vector has a high magnitude then it ought to be trusted? Note that this might be a reasonable assumption in the case of a pointer-generator style model, where a single attention vector is used both for attending and for copying, but in a model where attention isn't constrained in this way the magnitude of the attention vector may be misleading.

- The variational objective seems difficult to justify. First, I don't understand what it means for p(y | z, x) to be assumed to 1. Is this for any z (in which case y is independent of z)? Otherwise, how can it be removed from the objective? (Put another way: Equation (17) is not in general true; it neglects an expected log likelihood term). I'm also not entirely clear on how Equation (12) is modeled: do the z's really only rely on the other sampled z's? Doesn't that require a different model than the one that calculates P^{\kappa}?

- Somewhat minor: the claim that optimizing the joint objective needn't hurt perplexity relies on kappa being 0; have you confirmed empirically that when it isn't zero the perplexity improves over the baseline model?

- Finally, I'm not sure I understand why there needs to be a stop-gradient in equation (4). It would be nice to also verify empirically that this is important.



**Experience Assessment:**

I have published one or two papers in this area.

**Review Assessment: Checking Correctness Of Derivations And Theory:**

I assessed the sensibility of the derivations and theory.

**Review Assessment: Checking Correctness Of Experiments:**

I assessed the sensibility of the experiments.

**Review Assessment: Thoroughness In Paper Reading:**

I read the paper at least twice and used my best judgement in assessing the paper.

---

> ### Author Response · Authors · 2019-11-14
> **Response to Reviewer #4**
>
> Thanks for the detailed review. In addition to the revisions of our paper, we have also empirically investigated perplexity as suggested by the reviewer.
>
> > What justifies defining the attention score A_t in this way? Is it obvious that if the attention vector has a high magnitude then it ought to be trusted?
>
> The attention score A_t measures how much the model actually relies on the source to make a prediction. Since the context vector v_t is defined as a_t + h_t in Equation (2), defining A_t in this way as a magnitude ratio measures how much a_t affects v_t. In practice, A_t is usually high (~0.9) when the model is copying from the source (e.g. “Campbell” in Figure 2), medium (~0.4) when the model is generating based on some information from the source (e.g. “<unk>” in Figure 2), and low (~0.2) when the model is generating template elements (e.g. “is” in Figure 2). More examples are added to our revised paper. This observation does not immediately mean that a generation with higher attention score should be trusted; whether to trust a token or not is assessed by the confidence score. We will discuss this next.
>
> Our confidence score depends on both the attention score and generation probabilities. The idea of our confidence score was originated from an observation on baseline predictions of Wikibio: Hallucination often occurs when the table lacks some field that usually exists, for example the “Occupation” field is missing in Figure 1, and the baseline model makes that up. In such cases, the conditioned generation probability can still be high (e.g. >0.5), so we cannot tell it from the conditioned generation probability alone. But, since some usual field is missing, the attention score as we defined tends to be low, and it might be used to detect such hallucination cases. However, the attention score is also low for function words and template elements. Thus, we further incorporate an unconditioned generation probability (base-LM) to detect those cases.
>
> This motivation is made clearer in our revised paper.
>
> From a higher point of view, we do not claim that our definition of the confidence and attention scores are optimal; we proposed one way to implement the intuition. We have demonstrated several pieces of evidence that this implementation works: Figure 2 qualitatively demonstrates that the scores match human intuition; experiments regarding thresholding and ablation, etc. We believe they are all firm justifications for our model design.
>
> Regarding design details, we have tried several other variations of the attention score, the base-LM input-feeding, and the confidence score. The current implementation is selected based on PARENT F1 and manually investigating the learned scores.
>
> > what it means for p(y | z, x) to be assumed to 1
>
> Intuitively, we are assuming an oracle that can always recover the original target sequence y from its subsequence z; this is reasonable during training because we always know the gold reference for each training example. Technically, we note that p(y | z, x) is part of the model rather than the data. So this is just a modeling assumption that simplifies the formula. We assume p(y | z, x) to be a probability distribution over all possible target sequences, such that the probability is 1 for the gold reference and 0 otherwise. Yes, this distribution is the same for all z, as long as z is a subsequence taken from y. We don’t see any issue here.
>
> > how Equation (12) is modeled: do the z's really only rely on the other sampled z's?
>
> Yes.
>
> > Doesn't that require a different model than the one that calculates P^{\kappa}?
>
> No, we don’t need a different model. We are treating z as if z is the gold reference, and train our model to target this confident subsequence. This way, the generation model actually learns to skip unconfident tokens. (Reviewer #1 raised concerns about this setting that it might cause disfluent generations; the fact that it does not is also a surprise for us. Please see the discussions there.)
>
> > have you confirmed empirically that when it isn't zero the perplexity improves over the baseline model?
>
> This is an interesting question. First, we note an issue with perplexity on the WikiBio dataset: there are noisy tokens in the data whose log-likelihood converge to -inf. In our implementation, we set the smallest log-likelihood to log(2^{-100})=-69.3. Then, we compare the Pointer-Generator baseline with our No-variational model because the variational loss introduces random sampling into the training process. We report the log-perplexity as follows:
>
> No-variational, kappa learned: 31.19 (Train)  39.24 (Valid)
> No-variational, kappa=0:   31.21 (Train) 39.08 (Valid)
> Pointer-Generator:   32.41 (Train) 40.14 (Valid)
>
> So, compared to kappa=0, using learned kappa indeed improves training perplexity, but the validation perplexity gets worse. On the other hand, calibration improves the perplexity over the Pointer-Generator baseline, on both training and validation sets.

---

### Author Response · Authors · 2019-11-14
**Top level response**

We thank all the reviewers for their diligent work. In addition to addressing reviewer specific concerns below we have provided an overall list of changes here:

-Better explanations of our attention score and confidence score (as suggested by Reviewer #4 and Reviewer #3)
-Added a few related works (as suggested by Reviewer #2)
-Report number of annotators and annotation agreements in Section 5.1 (as suggested by Reviewer #3)
-Appendix 7.2 details our human evaluation template (as suggested by Reviewer #3)
-Appendix 7.3 gives examples where the baseline hallucinates and our approach remains faithful (as suggested by Reviewer #3 and Reviewer #1)
-Appendix 7.4 depicts the token-level attention score for several examples (as suggested by Reviewer #4 and Reviewer #1)

We will also be releasing our human annotations shortly and open sourcing the code after the review cycle.

Regarding evaluation (Reviewer #1 concern), we would like to highlight that: BLEU is a bad metric for measuring faithfulness/hallucination; and because of the length penalty, it is correlated with sentence length. Thus, we rely on PARENT precision (Dhingra et al. 2019) as well as a rigorous human evaluation to evaluate our models.

As a matter of fact, one can simply boost the BLEU score of the baseline Pointer-Generator model from 41 to 45 by setting Wu et al. (2016)’s length penalty to 2.0 at inference time; but this will hurt PARENT precision and decrease the faithfulness by 10 points according to our human evaluation.

---

### Comment · Area_Chair1 · 2019-11-14
**Reviewers, any comments on author response?**

Dear Reviewers, thanks for your thoughtful input on this submission!  The authors have now responded to your comments.  Please be sure to go through their replies and revisions.  If you have additional feedback or questions, it would be great to get them this week while the authors still have the opportunity to respond/revise further.  Thanks!

---

### Author Response · Authors · 2019-11-14
**Anonymous link to share human evaluation data**

Dear Reviewers,

We have shared our human evaluation results and model outputs via this link:
https://drive.google.com/open?id=1Kg4hJkaK9gWCv7mxwBfHEQwAgF_TrwcE
Please also refer to the other comments about our paper revisions and discussions.

Best,

---

### Decision · Program_Chairs · 2019-12-19

**Decision:**

Reject

**Comment:**

This paper proposes to improve the faithfulness of data-to-text generation models, through an attention-based confidence measure and a variational approach for learning the model.  There is some reviewer disagreement on this paper.  All agree that the problem is important and ideas interesting, while some reviewers feel that the methods are insufficiently justified and/or the results unconvincing.  In addition, there is not much technical novelty here from a machine learning perspective; the contribution is to a specific task.  Overall I think this paper would fit in much better in an NLP conference/journal.